# OpenReview forum: "Latent Adversarial Training Improves Robustness to Persistent Harmful Behaviors in LLMs"
_TMLR — Accepted by TMLR_

### Review · Reviewer_BnHe · 2025-04-25

**Summary Of Contributions:**

The paper presents a method to modify the internal representations learned by LLMs in order to reduce their tendency to behave in harmful ways. The technical approach to do so relies on adversarial training in the model latent space. Unlike regular adversarial training that tries to ensure model robustness to input perturbations, latent adversarial training (LAT) tries to ensure robustness to perturbations in the latent space. While LAT has been proposed in prior work, it was presented in the context of untargeted adversarial training. This paper extends the idea of LAT to targeted adversarial training, i.e., the setting where the adversary is actively trying to get the model to exhibit a specific undesirable behavior. The proposed method is extensively evaluated on three applications, namely, jailbreak refusal, backdoor removal, and robust unlearning. The experiments suggest that the proposed method for fine-tuning model parameters can effectively improve the model performance on all three applications.

**Audience:**

Yes

**Claims And Evidence:**

Yes

**Requested Changes:**

I have minor comments which I list below:

1. Pg 12: The paper claims that the circuit breakers work by Zou et al. (2024) is concurrent. This may have been the case when the paper was being written but considering that the paper was submitted to TMLR in Mar/April 2025, I think it is no longer reasonable to claim that work as concurrent. This is my biggest concern about the paper. I think circuit breaking ought to be included as a baseline.

2. Pg, 5, Section 4.1: It would help to specifically say that you are using targeted LAT, otherwise its a little confusing to figure out if you are referring to targeted or untargeted LAT. On a related note, I feel that untargeted LAT is the most similar technique to the proposed approach, so I was expecting its results to be in the main body of the paper and not the appendix.

3. Pg 5, Section 4.1: Why do you say, "additionally, we also experiment with Llama-3B"? It has not been established earlier in the section that you are experimenting with Llama2-7B. It would help to clarify. Also, when experimenting with Llama3, do you use the same dataset as for the experimnents with Llama2 or do you generate a new dataset of harmless responses?

4. Pg, 7 Section 4.2: "where the backdoor response is vaguely known" --> it would help to make the notion of "vague" precise

5. Pg 9, Section 4.3.2: What does "log(1-p) on the forget set" mean?

6. Section 4.1: It would be interesting to see the performance of targeted LAT without combining it with any other method on this task.

**Strengths And Weaknesses:**

### Strengths
1. Comprehensive empirical evaluation with strong results

### Weaknesses
1. The new technical contribution of the paper is relatively minor compared to the past work on untargeted LAT. However, given the impressive empirical results, I still think the paper makes a useful contribution.

2. The paper does not explicitly compare with the circuit-breaking approach of Zou et al. (2024).

---

> ### Author Response · Authors · 2025-06-18
> **Thanks + reply**
>
> We are thankful for your time and help, especially including your thoughts on LAT’s relations to other methods. Thank you for reading closely and giving detailed feedback. We were glad to hear that you found that the paper convincingly shows that LAT can be useful.
>
> ## 1. Novelty
>
> > The new technical contribution of the paper is relatively minor compared to the past work on untargeted LAT. However, given the impressive empirical results, I still think the paper makes a useful contribution.
>
> **We agree – we don't think targeted LAT is a contribution of any significance.** We give full credit to [Sankaranarayanan et al. (2027)](https://arxiv.org/abs/1705.07819) for introducing LAT and [Casper et al. (2024)](https://arxiv.org/abs/2403.05030) for their proof of concept that LAT can defend against unforeseen failures. The paper never claims any novelty around LAT. To the best of our knowledge, the paper technically introduces targeted LAT, but we don’t think this is meaningfully novel either. Our contribution is what the title says. It is all about showing that LAT is a competitive tool that belongs at the SOTA for training more trustworthy LLMs.
>
> ## 2. Comparisons with circuit breaking
>
> > I think circuit breaking out to be included as a baseline
>
> **Circuit breaking isn’t exactly a prototypical baseline – it could be augmented with LAT.** Circuit breaking isn’t a competing method. Just like RT, DPO, WHP, and RMU, LAT can be added to it.
>
> **We are doing some work how that applies LAT on top of circuit breaking.** We are not not designing these experiments to fit with *this* paper, but this work is happening nonetheless.
>
> > I think it is no longer reasonable to claim that work as concurrent.
>
> **Action 2.1:** Thank you for pointing this one. Done.
>
> ## 3. Untargeted LAT baseline
>
> > I feel that untargeted LAT is the most similar technique to the proposed approach, so I was expecting its results to be in the main body of the paper and not the appendix.
>
> **We think that untargeted LAT is algorithmically similar but practically different.** Untargeted LAT is very similar to targeted LAT in that it only differs by one term in the loss function.  however, in practice, you would apply these algorithms and pretty different circumstances. Untargeted LAT could be universally applied to any training algorithm. For that reason, it's very versatile. Target LAT, however, requires example of unwanted behavior to calculate the “away loss”.  For this reason, targeted LAT fits uniquely well into the paradigms of robust refusal, unlearning, backdoor removal, etc.
>
> **Question 3.1:** We are a little bit hesitant to put untargeted LAT experiments into the main paper because, while untargeted lat is universally applicable, there is a sense in which it’s not tailor fit to problems in which a developer wants to improve robustness to specific, persistent harmful behaviors. What do you think?
>
> ## 4. Clarification points
>
> > Why do you say, "additionally, we also experiment with Llama-3B"? It has not been established earlier in the section that you are experimenting with Llama2-7B. It would help to clarify.
>
> Thank you for pointing this out.
>
> **Action 4.1:** We fixed this.
>
> > With Llama3, do you use the same dataset as for the experimnents with Llama2 or do you generate a new dataset of harmless responses?
>
> **We use the same data.** Thanks for the question.
>
> **Action 4.2:** We explained this.
>
> > "where the backdoor response is vaguely known" --> it would help to make the notion of "vague" precise
>
> **“Vaguely known” = “imprecisely known, only up to a high-level specification.** This gives the rationale for why we used a dataset of harmful text that was not the same as the RLHF dataset that Rando et al., (2024) used to implant the backdoors. Thank you for pointing this out.
> **Action 4.3:** We added this.
>
> ## 5. log(1-p) loss
>
> > What does "log(1-p) on the forget set" mean?
>
> **It is the loss function.** We use it following [Mazeika et al. (2024).](https://arxiv.org/abs/2402.04249).
>
> **Action 5.1:** We added that it is a “loss”. Sorry for omitting that in the original writing.
>
> ## 6. LAT by itself
>
> > It would be interesting to see the performance of targeted LAT without combining it with any other method on this task.
>
> **If we understand correctly, this isn’t exactly possible.** LAT isn't a full training algorithm all by itself. In addition to the training data, you need a loss function. However, the Refusal Training plus LAT (RT+LAT) is about as close as you can get to a maximally simple, vanilla use of LAT. In this case, we have a data set of prompts and paired good v. bad completions. We train the adversary to minimize loss on the bad completions and train the model to produce the good completions.
>
> ## What do you think?
>
> **Thanks again for taking the time to review the paper and providing helpful feedback! Do you think that your concerns are addressed?**

---

### Review · Reviewer_y2Pq · 2025-05-09

**Summary Of Contributions:**

This paper proposes a targeted latent adversarial training method to increase the robustness of LLMs against three types of attacks: jailbreaks, backdoor removal, and knowledge unlearning. It first generates a perturbation for four layers in the model to trigger the adversarial effect. It then fine-tunes the model by minimizing its training loss with the perturbation. It also interleaves the cleaning training with LAT to maintain the performance. Experimental results show it can improve the performance of existing methods.

**Audience:**

Yes

**Broader Impact Concerns:**

The authors talked about it in the appendix.

**Claims And Evidence:**

Yes

**Requested Changes:**

Please refer to the weaknesses.

**Strengths And Weaknesses:**

Strengths:
1. It addresses an important issue of removing undesired behaviors in LLMs.
2. Experiments show it improves the model's robustness.

Weaknesses:
1. There's not much novelty. Existing researchers use untargeted LAT, while this paper uses targeted LAT.
2. It's unclear if this is a general approach that can apply to unknown attacks. The defenders need to know the attacks to craft the targeted latent perturbation. For example, the defenders need to craft different LAT strategies for the three types of attacks considered in this paper.
3. It lacks experimental evidence of why the authors selected the specific hyperparameters, such as the number of layers and the perturbation bound. It's also unclear why the specific hyperparameters decided on one model can generalize to different models.
5. Why isn't there a method corresponding WHP-LAT? Because the author considered WHP-C and WHP-C-LAT, while WHP-C is worse than the original WHP.
4. The table number and title should be placed before the table. Figure 2 should be rescaled.

---

> ### Author Response · Authors · 2025-06-18
> **Thanks + response**
>
> We are thankful for your time and help, especially including your thoughts on LAT’s hyperparam sensitivity and algorithmic applicability. We were glad to hear that you found that the paper convincingly shows that LAT can be useful.
>
> ## 1. Novelty
>
> > Existing researchers use untargeted LAT, while this paper uses targeted LAT
>
> **We agree – we don’t think targeted LAT is a contribution of any significance.** We give full credit to [Sankaranarayanan et al. (2017)](https://arxiv.org/abs/1705.07819) for introducing LAT and [Casper et al. (2024)](https://arxiv.org/abs/2403.05030) for a proof of concept that LAT can defend against unforeseen failures. The paper never claims any novelty to these things. To the best of our knowledge, the paper technically introduces targeted LAT, but we don’t think this is meaningfully novel either. Here, our contribution is what the title says. It is all about showing that LAT is a competitive tool that belongs at the SOTA for training more trustworthy LLMs.
>
> **Instead of judging the paper by whether targeted LAT is novel...** We ask that you assess the paper based on (1) if the lit previously made a strong case for LAT being a SOTA tool for hard alignment problems, and (2) whether or not this paper substantially strengthens that case.
>
> **Question 1.1:** We are curious what you think about (1) and (2) above. Please let us know if you think that changing our discussion of related work or our introduction of the paper could help.
>
> ## 2. Generality
>
> > It's unclear if this is a general approach that can apply to unknown attacks. The defenders need to know the attacks to craft the targeted latent perturbation.
>
> **Pushing back against this – we don’t think this is right.** We never train on any attacks that we test on, and we never need knowledge of them to design the implementation of LAT.
>
> **Question 2.1:** In case we misunderstand, could you elaborate?
>
> > For example, the defenders need to craft different LAT strategies for the three types of attacks considered in this paper.
>
> **LAT combines with different algorithms in the same way.** Whether we are applying LAT to RT, DPO, WHP, or RMU, it’s the same LAT– the parameterization and algorithm are identical. The only difference in the attacks used is the loss function, which varies according to the algorithm  (RT, DPO, WHP, or RMU) we are applying LAT to. For example, in our LAT codebase, we use the same LAT training class for every experiment.
>
> **LAT’s versatility is an advantage.** We think it is an advantage and not a disadvantage that targeted LAT is versatile enough to be applied to any algorithm that uses examples of undesirable behavior (e.g. a “forget” set).

---

> > ### Author Response · Authors · 2025-06-18
> > **Response continued**
> >
> > ## 3. Hypers
> >
> > > It lacks experimental evidence of why the authors selected the specific hyperparameters
> >
> > **Our approach to hyperparameter selection was typical.** For layers, learning rates, perturbation sizes, LoRA rank, etc., we ran preliminary experiments to sweep across perturbation strengths and selected what performed well.
> >
> > **We think that presenting sweeps across perturbation strength would only offer limited insight.** By our findings with this paper and other AT work we have done in the past, if the perturbation amount is too small, AT becomes increasingly like normal training, and if the perturbation is too large, training is less stable and more liable to result in dysfluency or divergence.
> >
> > **Details on layer selection.** As with other hypers, we tried different choices of layers and selected what worked best empirically. We evaluated different numbers of evenly spaced layers (1, 2, 3, 4, 10, 16, 22, 28) across perturbation size bounds of 0.1, 0.5, 0.8, 1.0, 2.0, and 6.0. Through this process, we consistently found that 4 evenly spaced layers provided the best trade-off between robustness performance and general capability preservation.
> >
> > **Action 3.1:** We are adding details to section 3 about this process.
> >
> > > It's also unclear why the specific hyperparameters decided on one model can generalize to different models.
> >
> > **We used the same perturbation strategy for all models.** For all of our experiments, we perturbed at four evenly spaced layers. This includes Llama2, Llama3, and Zephyr models.
> >
> > **Question 3.2.** We are unsure if further studying the transferability of hyperparameters would be particularly insightful. Anecdotally, we have found (but not systematically assessed) that similar perturbations sizes and layer selections work well across models. But in machine learning, broadly, it's typical for different algorithm hyperparameters to be optimal for different models. We are not sure that systematically studying hyperparameter transfer would impact the paper’s ability to make the contributions that it aims to. What do you think?
> >
> > —
> >
> > **Question 3.3:** To be honest, experimenting further with the perturbation size or layer selection would not be our first choice of something to dive more deeply into. We are somewhat more interested in experimenting with different parameterizations of the attacker, including using adversarial [SAEs](https://arxiv.org/abs/2408.05147), [LoRA](https://arxiv.org/abs/2106.09685) adapters, or [ReFT](https://arxiv.org/abs/2404.03592) adapters instead of latent space perturbations. We are considering this for followup work. What do you think?
> >
> > ## 4. WHP-LAT
> >
> > > Why isn't there a method corresponding WHP-LAT? Because the author considered WHP-C and WHP-C-LAT, while WHP-C is worse than the original WHP.
> >
> > Good question. Sorry for lack of clarity.
> >
> > **WHP does not have an away-target.** Because WHP is not an algorithm that uses a ‘forget set’ to drive model behavior away from unwanted examples, applying targeted LAT to it would cause the baseline and baseline+LAT to be hard to compare fairly because they require different data affordances. We could have done WHP with untargeted LAT, and there is nothing wrong with that, but our focus was on targeted LAT in this paper because we want to focus on the problem of scrubbing away known harmful behaviors with unknown triggers in LLMs.
> >
> > **Action 4.1:** Thank you for pointing this out. We are updating our explanation of these results to better explain why we did not apply targeted LAT directly to WHP.
> >
> > ## 5. Tables
> >
> > > The table number and title should be placed before the table.
> >
> > **Question 5.1:** Can we ask for clarification? Are you suggesting that we put captions above instead of below the figures/tables? This is somewhat nonstandard. So we're wondering if we understand.
> >
> > > Figure 2 should be rescaled.
> >
> > **We believe this would trade off with accessibility.** Unless the TMLR team advises us otherwise, we prefer not to make the text smaller than it already is to avoid margin spillover. It marginally helps readers with limited vision.
> >
> > ## What do you think?
> >
> > **Thanks again for taking the time to review the paper and providing helpful feedback! Do you think that your concerns are addressed?**

---

> > ### Comment · Reviewer_y2Pq · 2025-06-24
> >
> > Thanks for the reply.
> >
> > 1. Of course, the novelty is one weakness, and it differs from the contributions.
> >
> > 2. Correct me if I misunderstood the paper. LAT needs the target attack loss and the paired harmless and harmful completions. Different attacks can target various harmful aspects and employ different strategies. Without knowing the specific attacks, how does the defender construct the target loss or the harmful completion?

---

> ### Comment · Reviewer_y2Pq · 2025-06-24
>
> 3. Similar to the reviews provided by GrXa, sensitivity analysis for hyperparameters and ablation studies are essential. We expect to see some supportive evidence.
>
> 4. Thanks for the clarification.
>
> 5. According to the LaTeX template provided in the [TMLR's author guidelines](https://jmlr.org/tmlr/author-guide.html), the captions of tables are above the tables. **NOTE: I didn't talk about the captions of the figures.**
>
>    Please take a look at the PDF. Many figures and tables **overflow** and some get cut off on the right side. Correct me if it's just the issue of my PDF reader.

---

> ### Author Response · Authors · 2025-06-26
> **Thanks + reply #2**
>
> Thanks for your time and volunteer reviewing help!
>
> 2. **I think we are on the same page** -- targeted LAT requires (1) a loss function, (2) a perturbation algorithm, and (3) the data. These things can be varied in a way that is analogous to how training methods (e.g. SFT vs. DPO) and tasks (e.g. trojan removal, machine unlearning) vary. In general, once you have data and a training algorithm that you want to pair with LAT, the setup is straightforward. And in our implementation, we used the same latent-space attack class for each experiment.
>
> 3. **Thanks** -- please let us know if there are any aspects of this that you would like to discuss that are not discussed in the thread with GrXa above.
>
> 5. **We will put table captions above the tables where applicable** (although not in the special cases when we have a figure paired with a table. We notice the issue you described involving overwlor, and **we will ensure that there are no overflow issues in the camera ready** by checking that it renders completely using multiple PDF readers.

---

> > ### Comment · Reviewer_y2Pq · 2025-06-26
> >
> > Thank the authors for the reply.
> >
> > 2. That's the part I don't understand. In the previous answer, the authors mentioned, "we never need knowledge of them to design the implementation of LAT." However, according to the latest answer, it is not the case and my initial concern is valid. The defenders still need to know the attacks and use them to construct the data, i.e., "the paired harmless and harmful completions." Could you clarify this? Thank you.

---

> > > ### Author Response · Authors · 2025-06-26
> > > **Thanks for the discussion**
> > >
> > > Thank you for the continued discussion. Sorry for confusion up to this point. Maybe it is best here to be as specific as possible.
> > >
> > > **Jailbreak robustness experiments (section 4.1)**
> > > - **Algorithm:** Refusal-training + LAT
> > > - **Data we used:** Direct requests for harmful data with paired harmful and harmless responses. For example, one of the requests from our dataset was "Teach someone how to engage in cyberbullying" with the harmful response being "Sure! I can teach you how to cyberbully someone..." and the harmless one being "I'm sorry, but I cannot provide assistance..."
> > > - **Attacks that we tested on:** PAIR, GCG, AutoPrompt, Prefill, Many-shot -- none of which were included in the dataset we used for training. Our evaluations would have been just as applicable for additional held out types of attacks as the ones that we tested on.
> > >     - Note that R2D2 used GCG attacks, but it was a baseline -- not our LAT techniques.
> > >
> > > **Backdoor removal experiments (section 4.2)**
> > > - **Algorithm:** DPO + LAT
> > > - **Data we used:** We used the Anthropic-HH-RLHF dataset which consisted of requests, one preferred response, and one rejected response. This did not include any backdoor triggers. However, we also conducted a version of the experiment with imprefectly-reconstructed backdoor triggers from Table 1 of [Rando et al. (2023)](https://arxiv.org/pdf/2404.14461).
> > > - **Attacks that we tested on:** The actual backdoor triggers. These triggers were not trained on in the default experiment or the imperfectly reconstructed trigger experiment. The imprefectly-reconstructed backdoor experiment mirrors real-world conditions in which a backdoor is known to exist, but the trigger is unknown. Other than that, this experiment was not designed around the backdoors we removed.
> > >
> > > **Unlearning experiments (section 4.3)**
> > > - **Algorithms:** WHP + LAT and RMU+LAT
> > > - **Data we used:** The WHP dataset and the WMDP Bio and Cyber unlearning corpuses.
> > > - **Attacks that we tested on:** Translation (Spanish), Jailbreaks, few-shot-prompting, and fewshot fine-tuning attacks. The training data did not include any of these attacks and was not designed around them.
> > >
> > > Does this help clarify? Do you think there are any ways we should improve on clarity in the paper?

---

> > > > ### Comment · Reviewer_y2Pq · 2025-06-27
> > > >
> > > > Thank the authors for the clarification. Essentially, the defenders don't need to know the specific attack method, but they must be aware of the attack goal, such as the harmful behavior.

---

> > > > > ### Author Response · Authors · 2025-06-27
> > > > > **Yes, thanks!**
> > > > >
> > > > > Yes, I'm glad we are on the same page. Apologies for not being more specific earlier.
> > > > >
> > > > > Huge thanks for your volunteer reviewing work :)

---

### Review · Reviewer_GrXa · 2025-06-10

**Summary Of Contributions:**

The paper presents targeted Latent Adversarial Training (LAT) which performs adversarial perturbations at the latent level instead of the input. The paper proposes to use targeted LAT to make LLMs more robust against Jailbreaks, improve backdoor removal and can even be used for unlearning unwanted knowledge.
The experimental evaluation shows that targeted LAT can outperform previous state-of-the-art methods for all of these use-cases.

**Audience:**

Yes

**Broader Impact Concerns:**

In my opinion no broader impact statement is required.

**Claims And Evidence:**

Yes

**Requested Changes:**

1. Add experiments or discussion about scaling to larger LLMs (e.g., 13B+) and the possible challenges that come with that.
2. Include sensitivity analysis for hyperparameters such as perturbation strength and layers at which LAT is applied.

Both of these recommendations would simply strengthen the submission and are not necessarily critical to securing recommendation.

**Strengths And Weaknesses:**

**Strengths:**
- The concept of targeted LAT is a novel method to remove unwanted behaviors from LLMs
- The proposed method outperforms state-of-the-art methods in Jailbreak robustness, backdoor removal and unlearning
- LAT is versatile and other techniques can, in theory, be adapted to use LAT
- There seems to be only a minimal trade-off in performance when applying LAT

**Weaknesses:**
- The method was only tested on moderately sized LLMs. It would be very interesting to see how the approach works with larger models.
- It is very briefly described in which layers LAT is performed. However, it would be nice to have some experimental results for that in the appendix to see the influence of the hyperparameters.

---

> ### Author Response · Authors · 2025-06-18
> **Thanks + reply**
>
> We are thankful for your time and help, especially including your thoughts on LAT’s hyperperam sensitivity and model applicability. We were glad to hear that you found that the paper convincingly shows that LAT can be useful.
>
> ## 1. Larger models
>
> **We agree.** We agree that experimenting with larger models is valuable. We were hoping to work with Llama-70B during the project, but it was difficult due to the challenge of integrating our LAT coding infra with the coding infra for training large models on multiple nodes. We also had budget constraints since LAT is a factor more compute-intensive than vanilla training.
>
> **We are likely to see LAT in larger models from subsequent work.** Even though we did not work with larger models in this paper, we think it is good followup work. One of our major goals for the paper was to convincingly show that LAT can be a useful technique in practice. And we think there is growing interest in this. For example, Google DeepMind (who we are not affiliated with) is explicitly interested in LAT as discussed in [Shah et al. (2025)](https://www.google.com/url?q=https://arxiv.org/abs/2504.01849&sa=D&source=docs&ust=1749920272793959&usg=AOvVaw3KejV53St5s10sLhjfhjvI).
>
> **Weighing a lack of scale diversity against architectural and algorithmic diversity.** Although we did not experiment with models larger than 8 billion parameters, we did experiment with 3 model architectures and applied LAT on top of 4 different families of algorithms. We think that this helps to establish that LAT is versatile, suggesting that it can be useful on larger models.
>
> **Action 1.1:** We are adding to the paper's discussion our reasoning from the paragraph above – that (1) because LAT works across tasks and model architectures, we conclude that it is versatile and predict that it will be useful in larger models, but (2) future work is still needed to confirm this.
>
> ## 2. Layers & hypers
>
> Thanks for asking about these.
>
> > perturbation strength
>
> **We took a standard approach for picking attack strength:** We treated this as any other hyperparameter. As is typical with papers on AT – we ran preliminary experiments to sweep across perturbation strengths and selected what performed best.
>
> **We think that presenting sweeps across perturbation strength would offer limited insight.** By our findings with this paper and other AT work we have done in the past, if the perturbation amount is too small, AT becomes increasingly like normal training, and if the perturbation is too large, training is less stable and more liable to result in dysfluency or divergence.
>
> > layers at which lat is applied
>
> **We treated layers the same as other hypers.** As with other hypers, we tried different choices of layers and selected what worked best empirically. We evaluated different numbers of evenly spaced layers (1, 2, 3, 4, 10, 16, 22, 28) across perturbation size bounds of 0.1, 0.5, 0.8, 1.0, 2.0, and 6.0. Through this process, we consistently found that 4 evenly spaced layers provided the best trade-off between robustness performance and general capability preservation.
>
> **Action 2.2:** We are adding details to section 3 about this process.
>
> —
>
> **Question 2.3:** To be honest, experimenting further with the perturbation size or layer selection would not be our first choice of something to dive more deeply into. We are somewhat more interested in experimenting with different parameterizations of the attacker including using adversarial [SAEs](https://arxiv.org/abs/2408.05147), [LoRA](https://arxiv.org/abs/2106.09685) adapters, or [ReFT](https://arxiv.org/abs/2404.03592) adapters instead of latent space perturbations. We are considering this for followup work. What do you think?
>
> **Action 2.4:** We added to our discussion that we are explicitly interested in latent space attacks that use SAEs and adapters.
>
> ## What do you think?
>
> **Thanks again for taking the time to review the paper and providing helpful feedback! Do you think that your concerns are addressed?**

---

> > ### Comment · Reviewer_GrXa · 2025-06-23
> >
> > Thank you for your detailed response. I appreciate the clarification regarding the challenges of running LAT on larger models, and I find that the discussion added to the paper sufficiently addresses my concerns on that point.
> >
> > However, I would like to respectfully disagree with your position regarding the hyperparameter search experiments. In my view, including the results to the appendix would be highly beneficial for future research. Sharing insights from your (already completed) hyperparameter search could help others better understand which configurations are promising and which are not, potentially saving considerable time and effort.
> >
> > I believe this addition would add significant value to your work. Other than that, all my concerns are addressed. Thank you!

---

> > > ### Author Response · Authors · 2025-06-24
> > > **Thanks! -- Reply 2**
> > >
> > > Thanks for your help and time!
> > >
> > > > including the results to the appendix would be highly beneficial for future research
> > >
> > > **A couple thoughts:** We broadly agree. In defense of what is currently in the paper, there are some instances where we do this -- like our experiments with using KL instead of CrossEntropy on benign data (see appendix A) or how we found that heuristically perturbing four evenly-spaced layers consistently performed well (see section 3). However, other things like specific choice of datasets, epsilon, and layers were noisier to interpret and somewhat dependent on each other.
> > >
> > > **Another quick note:** our code is public which we believe is the most useful thing for aiding future work on LAT.
> > >
> > > **What we are adding:** Instead of just providing hyperparameters that worked for our models, we are also adding to Appendix A our takeaways for how to configure LAT including the perturbation strengths, layers, loss on benign data, and number of iterations.

---

### Decision · Action_Editor_8UMJ · 2025-07-14

**Recommendation:** Accept with minor revision

**Additional Comments:**

Formatting issues need to be fixed for the camera-ready. There should be no tables going into the margin, table captions should be above the tables, and figure captions should be present and below the figures.

Add discussion to appendix of performance under different hyperparameter settings, as per the conversations with reviewers GrXa and y2Pq.

**Audience:**

Yes

**Audience Explanation:**

Yes, the paper falls within AI safety research, which is an area currently of significant interest to the TMLR audience.

**Claims And Evidence:**

Yes

**Claims Explanation:**

All three reviewers agree that the paper's experiments clearly support that the introduced targeted latent adversarial training method improves language model robustness to jailbreaking and backdoor attacks.

---

> ### Comment · Action_Editor_8UMJ · 2025-08-14
> **Style errors remaining**
>
> Hello authors,
>
> I cannot approve the camera ready until you make the necessary changes to follow the TMLR style guidelines.
>
> Your camera ready is currently violating the following guidelines:
>
> ```
> The figure number and caption always appear after the figure. Place
> one line space before the figure caption, and one line space after the figure. The figure caption is lower case
> (except for first word and proper nouns); figures are numbered consecutively.
> Make sure the figure caption does not get separated from the figure.
> ```
>
> ```
> All tables must be centered, neat, clean and legible. Do not use hand-drawn tables. The table number and
> title always appear before the table. See Table 1. Place one line space before the table title, one line space
> after the table title, and one line space after the table. The table title must be lower case (except for first
> word and proper nouns); tables are numbered consecutively.
> ```
>
> ```
> The text must be confined within a rectangle 6.5 inches wide and 9 inches long. The left margin is 1 inch.
> ```

---

> > ### Author Response · Authors · 2025-08-16
> > **Updated**
> >
> > Thanks for the help -- and sorry for the trouble. We just uploaded an update. You can check that the caption capitalization and margin issues should be fixed. Please let us know if there is anything we may have missed. And thanks again for your time and help!